# A Regulatory View on Smart City Services

**DOI:** 10.3390/s19020415

**Published:** 2019-01-21

**Authors:** Mario Weber, Ivana Podnar Žarko

**Affiliations:** 1Sedam IT d.o.o., HR-10000 Zagreb, Croatia; 2Faculty of Electrical Engineering and Computing, University of Zagreb, HR-10000 Zagreb, Croatia; ivana.podnar@fer.hr

**Keywords:** smart city, regulatory characteristics, taxonomy, internet of things, interoperability

## Abstract

Even though various commercial Smart City solutions are widely available on the market, we are still witnessing their rather limited adoption, where solutions are typically bound to specific verticals or remain in pilot stages. In this paper we argue that the lack of a Smart City regulatory framework is one of the major obstacles for a wider adoption of Smart City services in practice. Such framework should be accompanied by examples of good practice which stress the necessity of adopting interoperable Smart City services. Development and deployment of Smart City services can incur significant costs to cities, service providers and sensor manufacturers, and thus it is vital to adjust national legislation to ensure legal certainty to all stakeholders, and at the same time to protect interests of the citizens and the state. Additionally, due to a vast number of heterogeneous devices and Smart City services, both existing and future, their interoperability becomes vital for service replicability and massive deployment leading to digital transformation of future cities. The paper provides a classification of technical and regulatory characteristics of IoT services for Smart Cities which are mapped to corresponding roles in the IoT value chain. Four example use cases are chosen—Smart Parking, Smart Metering, Smart Street Lighting and Mobile Crowd Sensing—to showcase the legal implications relevant to each service. Based on the analysis, we propose a set of recommendations for each role in the value chain related to regulatory requirements of the aforementioned Smart City services. The analysis and recommendations serve as examples of good practice in hope that they will facilitate a wider adoption and longevity of IoT-based Smart City services.

## 1. Introduction

In the last decade cities around the world are investing considerable effort into transformation towards the so called smart or intelligent cities, i.e., cities that are self-sustainable and understand the importance of its physical and digital infrastructure. Cities are investing in their infrastructure primarily to improve the performance of relevant city services [1], since it is well known, even from the Roman times, that investment in infrastructure (e.g., roads in Roman times) results with economic growth. In addition, with the introduction of new or even upgraded infrastructure based on Information and Communication Technology (ICT), cities are becoming more efficient, sustainable and friendlier to citizens by improving their quality of life in all aspects. These positive changes can be seen in economy, governance, mobility, environment and living [2].

While analyzing relevant literature and existing Smart City projects, we came to the conclusion that there are still gaps related to the analysis of regulatory aspects of Smart City services, their interoperability requirements and relevant best practices. In this paper we introduce taxonomy of Smart City service characteristics, and present four characteristic examples of Smart City services to identify their regulatory and technical characteristics in accordance with the proposed taxonomy. Based on this analysis, we have identified the major technical and regulatory requirements for the most common classes of Smart City services. These characteristics can be used by Smart City planners and various stakeholders involved in provisioning of such services to easily identify the vital technical and regulatory aspects they need to focus on while planning, designing, implementing and deploying Smart City services.

After defining regulatory and technical characteristics of Smart City services, we also identify roles in a Smart City ecosystem, or better to say, we define who is involved in design, implementation and provisioning of Smart City services and how the stakeholders relate to each other. From device provider to end user, there are many roles with specific contribution to provisioning of a specific Smart City service. Their interconnection, separation of concerns and clear rules are necessary for a smooth deployment and operation of deployed services.

Many cities around the world have started their digital transformation. In the EU, a series of EU-funded research projects has created a strong initial push to this process with a number of innovations and examples of good practice which are now spreading across Europe. One of the first and well known examples is the SmartSantander project (http://www.smartsantander.eu/) which has served as an initial lighthouse project for the following ones, Organicity (https://organicity.eu/) and Open & Agile Smart Cities (https://oascities.org/). Synchronicity (https://synchronicity-iot.eu/) is a running large scale pilot project which integrates many citywide data sources by means of the FIWARE software components. However, the aforementioned projects are manly oriented towards technical aspects and scaling of Smart City pilots, while regulatory aspects are not widely covered. In the US, the White House has recognized that “an emerging community of civic leaders, data scientists, technologists, and companies are joining forces to build “Smart Cities”—communities that are building an infrastructure to continuously improve the collection, aggregation, and use of data to improve the life of their residents” [3], and in 2016 the President’s Council of Advisors on Science and Technology called for the development of a Smart City platform and referred to it as “the City Web” where all relevant stakeholders share results, insights and best practices [4]. Although the aforementioned projects and initiatives represent notable examples of successful Smart City project, we believe that general regulatory aspects of Smart City services need additional analysis and classification to pave the way for a widespread use of such services in many different cities.

Although there are many papers on Smart City services and their technical realizations, to our knowledge there are few papers dealing with regulatory issues and legal obligations of stakeholders involved in a Smart City ecosystems. In [5] the authors are dealing with the regulatory framework for data protection introduced through the new EC’s General Data Protection Regulation (GDPR) and are trying to give an answer to a question whether this new regulatory obligation will slow down Smart City deployments or not. GDPR is usually related to privacy which was introduced in [6] as a way to “systematize the application areas, enabling technologies, privacy types, attackers, and data sources for the attacks, giving structure to the fuzzy term “Smart City” to identify privacy threats and possible answers to those threats. The authors have investigated new Smart City standards in [7]; however, policy regulations were not covered in this paper.

The first article introducing taxonomy of Smart City services was given by Benjelloun et al. [8] with focus on logistic projects, without looking into other aspects of a Smart City. In their article on new taxonomy of Smart City projects [9], the authors propose a classification covering different aspects of Smart City projects. They also identify a limited taxonomy for objectives, tools and stakeholders, but it is not adequate for regulatory aspects that are extensively described in this article. The work closest to ours is reported by the EU Smart Cities Information System (SCIS) initiative in [10]. The document lists and identifies specific “worst practice” scenarios which occurred due to limited local, regional and national policies which can impede successful projects. It is thus complementary to our paper which identifies general regulatory rules that need to be taken into account in early stages of Smart City projects.

The main contribution of this paper is the proposed Smart City taxonomy that identifies relevant technical and regulatory characteristics of Smart City services. The taxonomy is used to analyze representative examples of real Smart City services that have been chosen as representative Smart City services due to their well adoption in practice and/or interesting technical aspects. One of the major contributions is a set of regulatory recommendations for each role in the IoT value chain. Recommendations can serve as a valuable support to stakeholders during the process of planning and designing their Smart City solutions. By using these recommendations, a stakeholder can easily identify its responsibilities and focus on vital aspects thus shortening service development cycles. Our goal is to provide guidelines and examples of good practices which can lead to successful and widespread replicability as well as massive deployment of Smart City solutions.

The paper is organized as follows: Section 2 presents the whole ecosystem of Smart City services and identifies the roles in Smart City projects. Section 3 provides a detailed description of technical and regulatory characteristics of Smart City Services, while Section 4 introduces the proposed Smart City taxonomy with a set of recommendations for all stakeholders involved in the Smart City value chain and described in presented use cases.

## 2. An Ecosystem of Smart City Services

Among many Smart City definitions, we put forward the following two specified by standardization bodies since they are both precise and comprehensive. The ITU-T Focus Group on Smart Sustainable Cities analyzed nearly 100 definitions and used them to conclude the following: “A smart sustainable city is an innovative city that uses information and communication technologies (ICTs) and other means to improve quality of life, efficiency of urban operation and services, and competitiveness, while ensuring that it meets the needs of present and future generations with respect to economic, social and environmental aspects“ [11]. Another definition that can be seen as the most comprehensive one is given by ISO where Smart City is recognized as “a new concept and a new model, which applies the new generation of information technologies, such as the internet of things, cloud computing, big data and space/geographical information integration, to facilitate the planning, construction, management and smart services of cities“ [12].

As a completely new concept, Smart City is much more than the simple usage of technology to facilitate city services, but rather an overall strategy specific to each city that clearly identifies city’s strategic goals and defines practical guidelines on how to achieve these goals. Smart Cities require a completely new management approach to their infrastructure and services as well as novel communication mechanisms with its citizens. A continuous and iterative process should be in place to identify, deploy and offer new citizen-oriented services, while citizens are placed in the loop to influence the future of both deployed and forthcoming services. In such a dynamic environment, pilot projects are typically used to test the benefits of digital transformation in practice.

A typical pioneering example of a Smart City service is Smart Metering which uses smart meters for in-house measurement of electricity consumption, while these measurements can be accessed remotely to offer real time information about household energy consumption. Another popular example is Smart Parking which is usually one of the first services to be deployed in a city since it can significantly reduce traffic jams. There are many other examples, e.g., water leakage detection, intelligent transportation, smart street lamps, or air pollution detection. Many such services are available and deployed today in a number of cities around the globe. In addition, novel examples of “lifestyle” services are also emerging which do not require specific IoT devices, sensors or other networked city infrastructure, but are targeting citizens and their living habits. Of course, lifestyle services require the usage of smartphones and mobile applications. For example, Bologna has recently introduced a new application to incentivize citizens to use public transportation, bikes, or to simply walk (Bella Mossa and the Better Points application, for more information visit https://www.bellamossa.it/). The application tracks citizen activities to generate points which can be exchanged for vouchers, e.g., goods and services at local businesses. This application also falls within the category of Smart City applications since it uses ICT to accomplish Bologna’s strategic goal of reducing the number of cars in its city center.

An ecosystem of Smart City services does not refer just to a city itself, but also to everything that the city and its surroundings involve, like citizens with their social activities, infrastructure, and technology. It is a broad term that exceeds city boundaries and evolves with every new service, new set of networked devices, new infrastructure or even old infrastructure which is used in new ways since it is “powered by ICT”. By introducing Internet of Things (IoT) technologies, cities are increasingly becoming a living organism that can respond to the needs of citizens, but can also correct themselves if needed by learning from the environment on how to react in different situations based on experience. Note that cities already own and operate a huge infrastructure which can become “networked” by the use of ICT and IoT in particular.

An ecosystem of Smart City services consists of services built using the IoT stack. A typical high-level IoT stack is always layered and starts from "things" at the bottom of the stack where various devices, sensors, actuators and gateways are deployed in local spaces. On top of things there is a network layer providing connectivity of things to IoT platforms. IoT platforms are typically deployed within cloud infrastructure and serve as virtual representations of real things, e.g., they offer services to retrieve raw readings generated by networked sensors or they implement specific primitives to trigger actuation functions on specific networked devices. IoT platforms are also responsible for the management of deployed things, and serve as sources of raw data for data analytics and/or provisioning of higher-level services offered to end users. Following this technologically complex architecture which requires different skills for implementation, deployment and operational maintenance, the realization of an IoT-based Smart City service requires different stakeholders to assume specific roles within each of the previously listed IoT stack layers. Therefore, multiple providers in a partnership relation are typically involved in the provisioning of a single service. Figure 1 depicts the identified stakeholders and an IoT value chain model for Smart Cities that extends the initial value chain originally introduced in [13].

We can identify four main roles within the IoT value chain:Infrastructure provider: provides IoT devices and infrastructure which is connected to the Internet. In case when a telecom network is used to provide connectivity, an infrastructure provider is a provider of electronic communications services in accordance with the Electronic Communications Act. An infrastructure provider offers network connectivity (wireline or wireless) that is used to connect deployed IoT devices to the Internet. This network has to be built, installed and maintained in order to ensure continuous device availability and connectivity. It is a service that is normally provided for a fee and consists entirely or partially of signal transmission via electronic communication networks.IoT platform provider: offers an IoT platform and system functions that deliver easy access to sensor data and managed devices as well as data integration for development of new services. It is responsible for controlling heterogeneous devices and collecting data from various sensors. The usage of standard protocols is key to the success of his/her business model.IoT service integrator: a provider of IoT-related Smart City services which are built on top of one or a number of IoT platforms and add value to underlying platform services.IoT user: a buyer of a Smart City service that uses novel networked components in his/her products (e.g., a smart meter) and/or provides innovative services (e.g., a Smart Metering service) to end users. For example, an electricity company uses smart meters to improve their existing product and offerings by replacing existing devices with new ones that offer completely new services to citizens: e.g., real-time information about their energy consumption. IoT users interact either directly with infrastructure providers and use the provided low-level services within their infrastructure, or with IoT service integrators who offer bundled high-level services on top of IoT platforms.End user: a user at the end of the IoT value chain that buys and/or uses a Smart City service. An end user may be a private person or a company.

With a more detailed segmentation of stakeholder roles, we can also identify the potential for further granulation of IoT roles which increase the market competitiveness, and certainly open space for development of new companies and business models:Device provider is a manufacturer and/or reseller of sensor devices. He/she is engaged in the development, implementation and maintenance of the physical smart infrastructure placed in the city environment. Device provider offers sensors, actuators, gateways, and other networked devices which can be either directly connected to the Internet or might create wireless sensor networks with data aggregation capabilities as well as local smart spaces.IoT connectivity provider: offers device connectivity to the Internet, potentially using low-power wide area (LPWA) technologies. The connectivity can be accomplished using either the unlicensed (e.g., Zigbee, LoRa or SigFox) or licensed spectrum (e.g. NB-IoT, 5G, LTE, LTE-A). It is necessary to distinguish services based on connectivity since it defines the set of rules and regulations governing the provided IoT services. If unlicensed spectrum is used to connect all devices which are used to provide a specific Smart City service, the end service falls within the category of Internet society services and is not regulated by a National Regulatory Authority (NRA). In case licensed spectrum is used for device connectivity, all provided services have to follow the strict rules of regulated electronic communication framework and national and/or EU legislation. Further on in this paper these services are named non-regulated and regulated services, respectively. It becomes evident that such a distinction between non-regulated and regulated services may be difficult to establish in complex environments with many heterogeneous devices and multiple IoT platforms forming an ecosystem of Smart City services.IoT application and service developer: offers the development of IoT services and applications (typically mobile and web applications) based on available devices and platforms. IoT application and service developers build products for both IoT users and end users, i.e., citizens.

Let us consider a concrete example of a Smart Parking service to explain the proposed IoT value chain and specific roles within the value chain. The City of Split is the second largest city in the Republic of Croatia and the biggest city on the Croatian Adriatic coast with almost 200,000 inhabitants as well as the main passenger harbor. Split is one of the oldest cities in Croatia and its heart is permeated with narrow streets that are not constructed for a large number of vehicles. The number of vehicles triples during summer time, which causes severe parking problems and traffic congestions. Parking places are managed by a city company called SplitParking responsible for managing, maintaining and charging of parking spaces. SplitParking represents an IoT user that enhances its existing parking spaces with additional IoT devices providing information about parking occupancy, while its traditional parking service is enhanced by a mobile application guiding cars to free parking spaces (Split Parking application, for more information and links to available mobile applications visit http://www.splitparking.hr/smart-splitparking). SplitParking requires an infrastructure provider to place sensing devices within its parking places to monitor space occupancy. A device provider is the manufacturer of sensors for parking places with responsibility to ensure seamless integration of different types of adequate sensors with an IoT platform. To fulfill this, a device should support open protocols, while the device provider needs to ensure data integrity and security during communication between a device and the platform. An infrastructure provider has a two-fold role in this use case. The first role is that of a network provider to connect devices to the Internet and IoT platform, while the second role relates to connectivity between IoT devices and IoT platform that can be achieved through a contract with an electronic communication operator. The infrastructure provider must ensure lawful interception to open interfaces toward responsible legal entities. An IoT Platform provider ensures a software platform for managing devices and acquires data stemming from those devices so that adequate and up-to-date information is available to an IoT service integrator offering an IoT-enhanced parking service to SplitParking. SplitParking would preferably interact only with an IoT service integrator offering a convenient parking service integrated within a user-friendly mobile application. However, since SplitParking is a service provider to end users, it is primarily responsible for the privacy of end user’s data, and as an owner of parking spaces, it is also responsible for all other regulatory and technical challenges of the provided Smart Parking service. Thus, SplitParking needs to enter into a partnership with an infrastructure provider, IoT platform provider and IoT integrator to offer a new Smart Parking application and service to citizens and tourists.

Deployed Smart City services typically use proprietary solutions that are not compatible with each other and may as well be implemented as closed solutions. This creates high fragmentation that has been identified in [14] as one of the biggest challenges for the growth of the Smart City ecosystem. To overcome the traditional silo based organization of the cities, where each utility is responsible for their own functionalities, interoperable solutions are needed and although it’s not strictly connected to technology, resolving this barrier also requires change in the mindset of the city employees, or organizational changes. Interoperability, as defined by the ETSI’s Technical Committee TISPAN [15], states that interoperability is the ability of equipment from different manufactures (or different systems) to communicate with each other within the same infrastructure (same system). The aforementioned ETSI Whitepaper identifies different interoperability aspects (technical, syntactic, semantic and organizational). In Smart City context, syntactic and semantic are the most prominent interoperability aspects. Syntactic refers to the usage of open and standardized protocols and data formats, while semantic relates to solutions and schemes to describe information created by sensors in different applications to enable useful exchange of information between different city services. Existing solutions are typically based on proprietary data models that are not understood and recognized by other services, and the process of integration of devices and data across domains and IoT solutions becomes burdensome.

A key element to achieve interoperability in such environments with many existing systems and heterogeneous devices and information models is the usage of a horizontal platform for loosely-coupled integration of vertical solutions. This approach is proposed in [16] to enable interoperability of Smart City verticals: The authors presents a new approach that “consists in the definition of a set of modular, general specification (the Smart City Platform Specification) for implementing horizontal ICT platforms, in order to enable interoperability among the vertical silos”. A comparable approach is perused by the H2020 project symbIoTe [17] which implements a unique middleware framework for IoT interoperability that is published as open source software. The symbIoTe middleware extends existing IoT platforms and devices with specific adaptors that implement features relevant to semantic interoperability and attribute-based access control to offer unified REST-based interfaces on top of platforms and devices. Further information is available in [18].

## 3. Smart City services: an Overview of Technical and Regulatory Characteristics

A vast number of different IoT services is used in different sectors, from environment to government sector, and from people centric to technology oriented services. Currently vertical standalone IoT solutions prevail on the market: such solutions are restricted to an ecosystem that can be created around a single IoT platform [19]. Such vertical solutions do not share infrastructure nor generated data with each other, although the same infrastructure and data could be used by and incorporated into different services (e.g., temperature and NO_2_ concentration is important in an air quality monitoring system but also for calculation of green routes for bicycles or park irrigation systems). Therefore, a classification of services according to their characteristics is necessary to understand both their technical and regulatory requirements. In this section we identify technical and regulatory characteristics that need to be well-defined if we want IoT services to be successful in Smart City context as well as to fulfill the legal and technical requirements that are currently put in place or could be set as regulatory obligations in near future. Based on an extensive analysis of Smart City services we distinguish and classify technical and regulatory characteristics.

### 3.1. Technical characteristics of Smart City services

Technical characteristics of Smart City services are mostly related to Quality of Service (QoS) and other technical parameters that are strictly related to service infrastructure and design. We make a distinction between basic service characteristics and IoT Device characteristics. The basic characteristics relate to technical and QoS parameters which are common to all Smart City services and include the following:Number of end users: number of users that will use a service.Number of IoT devices: number of devices that will generate sensor data or be used as actuators.Data volume: the entire volume of generated data per service which in turn generates traffic to servers hosting IoT platforms. For most Smart City services the generated data volume is low; however, there are some specific services, e.g., video surveillance, that potentially generate large quantities of data.Time sensitivity: some of the services are sensitive to latency, like e-health services, where it is important to react immediately to specific events and therefore real-time provision and efficient data processing is necessary for such services, e.g., medical staff needs to react on time.Location-based service: if a service depends on location, then it is necessary to foresee such possibility and identify whether outdoor or indoor positioning is needed. Next, an appropriate positioning method needs to be identified offering adequate precision for the designed service. For example, in case of Smart Parking, precise outdoor location of a car is necessary to identify an appropriate parking lot.Billing: a service may be free of charge or in some scenarios end users would pay for a service (monthly subscription, pay per use, etc.);Scalability: most of the cities are starting with pilots on a smaller city area, and thus it is necessary to design the software architecture for future expansions in terms of increasing number of devices and users.Platform openness and interoperability: IoT platforms should expose unified and simple interfaces so that various innovative services and applications may integrate IoT devices and corresponding data managed by different IoT platforms. This will indeed create an ecosystem of interoperable Smart City services which reuse existing open infrastructure in new ways and can lead to a higher adoption of Smart City services by citizens. Cities need to demand open and interoperable infrastructure and software solutions from suppliers of devices and software since only interoperable IoT solutions can easily be integrated into novel innovative cross-platform and cross-domain services and applications.

IoT device characteristics are the following:Communication mode: when designing a city-wide solution, a city planner should take into account the required communication infrastructure to ensure connectivity of IoT devices to the Internet, while also taking into account particular geographical characteristics of an area. Adequate wired or wireless communication protocols should be used. There are many competing protocols and communication technologies on the market today, and for the future use at least a number of standardized options should be supported.Computing capability: this feature can dramatically change service operations and logic. In case that a device has sufficient computing capability, some operations and simple algorithms can be performed on the device, while reducing communication with the server or platform. However, in addition to an increased price, a device will also consume more power for computing. Therefore, it should be carefully evaluated whether for a specific service additional computing capability should be used on a device or whether the computation will be done within the cloud or at network edge.Power consumption: one of the crucial characteristics of an IoT device. Today devices are mostly battery powered and thanks to better efficiency of devices and their low power mode of operation as well as improved battery capacities, device longevity is largely improved. However, some devices still have to be connected to main power due to their consumption characteristics. Lately we are also experiencing the rise of energy-harvesting techniques combined with low-power devices [20].Power source: a device can be battery or main powered, or may employ an energy-harvesting technique.Location: if used for a location-based service, device location needs to be known which typically requires an additional GPS chipset to be included into a deployed device.

Connectivity-related characteristics of an IoT device include the following:One-way/two-way communication: declares whether a one or two-way communication is needed between device and server hosting an IoT platform.Bandwidth: it is important to estimate the rate of data transfer between a device and IoT platform based on service requirements (e.g., video surveillance needs high bandwidth compared to Smart Metering);Delay: some services are sensitive to delay in information delivery; for instance, real-time applications such as Smart Parking are highly sensitive to network delay.Jitter might not influence most services; however, for some specific services its influence may become significant.Loss of data: some services may be highly sensitive to data loss and thus special mechanisms should be in place, e.g., retransmission, to protect from data loss. However, some services might function correctly and be designed to be resilient to data loss.

Service designers need to analyze all of the listed technical characteristics of their services and devices so as to specify the requirements of a future Smart City service.

### 3.2. Regulatory characteristics of Smart City services

Regulatory characteristics are related to legal acts and bylaws relevant to Smart City services. They are predefined by the national or EU directives applicable to all EU member states. The most critical characteristics of a service relate to the following eight (8) characteristics:Lawful interception: in many countries national legislation requires the possibility of data traffic interception which thus also applies to IoT traffic.Service dependability: ability to avoid service failures that are more frequent and more severe than acceptable.Personal data protection: one of the fundamental human rights which stipulates that citizens have the right to protect their personal data.Security: concepts and solutions preventing cyber-attacks at the device and service level.Operator switch (number portability): ability to change an IoT operator, i.e., any stakeholder from the IoT value chain.Roaming: relevant to regulated services where foreign numbers are used outside of their domestic network, i.e., IoT devices registered in one network are used in visited networks.Interoperability and open access to data and services: we believe this is not only a technical, but also regulatory requirement, primarily for all publically funded Smart City services. It is highly related to operator switch.

National legislation in many countries implies a need to enable lawful interception for data traffic, phone calls and SMS/MMS messages for national security purposes. However, since IoT architecture allows to connect devices all over the world, it is an open question who can and is allowed to intercept IoT data. Officially, a service is located in one country, usually where its server is placed, however, devices could be placed in several countries, or devices can be placed in trucks that are driving through several countries. Additionally, if the service is placed in one country and service provider is a multinational company, in most cases SIM cards issued by a single country will be used by all members of this company or special IoT/M2M SIM cards could be used. As an example, if a service is offered in Croatia using a foreign or global subscriber number, the Croatian national authority for lawful interception will not be able to obtain an IP address of a device since the user device uses an IP address from a GGSN of an Austrian network. This represents a huge problem for national security. Usage of foreign numbers outside of a domestic country represents a roaming service. Roaming has created great challenges for service providers in Europe, however, the European Commission (EC) has decided that for the prosperity of the EU and its citizens, all EU countries should act as one market, therefore EC has delivered the Digital Single Market strategy for Europe [21]. Within this strategy EC introduced the “roam like at home” rules. This means that when one is using a mobile phone while travelling outside his/her home country in any other EU country, he/she does not have to pay any additional roaming charges. All citizens in EU have benefits from these rules when calling, sending text messages or using data services while abroad. This roaming regulation [22] has also facilitated the development of IoT services since there are no roaming charges implied anymore, however, there are some fair user policy limitations which are introduced in REGULATION (EU) 2016/2286 [23], e.g., a limitation that a device has to spend more time at home then abroad. For IoT services at a fixed location this means that multinational operators cannot use one numbering range for all deployed IoT devices in various countries, but rather have to use different ones for each country or need to use IoT/M2M SIM cards which are in principle more expensive. In this context, service providers should pay attention to the nature of a service when designing it to choose an appropriate communication channel so that roaming regulation rules do not restrict their service.

Service dependability is closely related to security and represents the ability to deliver a service that can justifiably be trusted or, in other words, dependability of a system is the ability to avoid service failures that are more frequent and more severe than is acceptable. Dependability often comes hand in hand with security since cyber-attacks as well as service and device vulnerabilities are frequent causes of service failures, and thus the most common attributes that it encompasses are availability, reliability, safety, integrity, maintainability and confidentiality. In this context system availability represents readiness for correct service provision; reliability relates to continuity of a service which functions without failures; safety points to the absence of serious or catastrophic consequences for the user(s) and environment; integrity is absence of improper system alteration and relates to service accuracy and consistency; maintainability is the ability of a service to undergo modifications and repairs; confidentiality is the absence of unauthorized disclosure of information.

One characteristic that could significantly influence Smart City service deployments over time is the change of operator which does not relate only to provider of electronic communication network, but also to IoT platform provider and even IoT integrator. If stakeholders use proprietary and closed solutions, this creates switching barriers and negative consequences, usually known as the “vendor lock-in” problem since technically it is quite difficult to replace such solutions without additional investment. However, the change of operator should be enabled, and is even a requirement for regulated electronic communication services, but it has so far not been stressed as a vital requirement for Smart City service deployments. This requirement is another strong argument in favor of standardized, open and interoperable solutions which have to be deployed at all layers of the IoT stack for Smart City services. Open access to data and services is closely related to interoperability. To foster deployment of Smart City services, it is necessary to make services interoperable, especially in cities that have already started their digital transformation. Open access on the other hand can provide data to anyone interested, thus resulting in many interesting incentives and ideas for novel Smart City services. We can see great examples of European cities such as Barcelona, Santander and Amsterdam which create vibrant ecosystems of innovative citizen applications based on open data.

Following the GDPR that was put in force on 25th of May 2018 and e-privacy directive that, according to the EC, should be available soon, great effort will be put in personal data protection. Both of these legal regulations are based on information-related security characteristics known as CIA (confidentiality, integrity and availability). These directives stress a requirement to expose existing infrastructure in cities and open up their data to citizens. Since in practice the infrastructure is mostly deployed and owned by cities, the problem relates to collected data which is stored by IoT service integrators who may claim to own this data. In addition, citizens can generate a substantial amount of data about their environment using Mobile Crowd Sensing (MCS) services. The question arises on who owns this IoT data? Who has the right to use it and to access the infrastructure from the actors in the value chain? For example, in MCS for air quality monitoring where individuals with sensing devices and smartphones share data about air quality for a specific micro location, the question is who is the owner of this data: citizens, city administration, IoT platform provider or someone else? Consider another example where a driver is speeding and this is detected by a built-in car sensor. What should the car do in this scenario? Should this data be sent to police department since the speed limit has been exceeded or this data belongs to a car owner and he/she should decide what to do with this data? Perhaps this data belongs to the car manufacturer hosting the IoT platform? According to GDPR, this data represents private information since it can be used to identify a person, and therefore we believe this information should belong to the car owner. On the other side, a new regulation could be defined which requires a car owner to share all relevant personal data with the authorities, police in this case, e.g., during the regular yearly car inspection. In this particular case personal data would need to be shared with a third party.

The previous discussion only scratches the surface of regulatory issues and possible questions that might arise in future. Thus it is vital that all of the listed regulatory characteristics are carefully analyzed by all stakeholders of the IoT value chain during the initial planning of a Smart City service. In addition, local regulations (laws and bylaws) may represent another burden for a massive deployment of IoT solutions and create obstacles for service replicability, as identified in a recent report by the EU Smart Cities Information System (SCIS) [10] where a number of real-world examples are reported. Recommendations are provided to local and national governments on what needs to be improved to remove these obstacles so that service replicability is enabled. To facilitate replicability and eventually massive deployment of IoT services, we believe that lightweight regulation with a monitoring process is needed in early phases of service adoption, while for specific services which are identified to require regulation, such regulation should be adopted on a large scale (e.g., at EU level or even world-wide). A good example is the Pan-European service e-call that in addition to EU countries has also been adopted by other non-EU countries (e.g. Russia).

### 3.3. Taxonomy of Smart City service characteristics

Based on the previously identified technical and regulatory characteristics, we summarize the taxonomy of Smart City service characteristics and depict it in Figure 2. The proposed taxonomy provides a classification of major characteristics, both technical and regulatory, which are vital to be determined for any Smart City service during an initial phase of service requirements specification. Thus, it can guide Smart City stakeholders through the process of identifying key characteristics of their future services in accordance with service technical and regulatory requirements. In the next section the taxonomy is used to characterize and identify specific requirements for several Smart City services, namely, Smart Metering, Smart Parking, Smart Street Lighting and MCS.

## 4. Selected Use Cases and Recommendations

### 4.1. Smart Metering

One of the EC’s objectives is to publish “the energy service directive” which expects that 80% of the households in EU will install smart energy meters by 2020 [24]. Smart Metering services are used to measure electricity consumption in real time so that a consumer can monitor it using a dedicated application. This information is later used by an electricity provider that can charge for consumed units, or may offer dynamic change of tariffs. Obviously, the electricity operator will have to process personal data to be able to offer such services. According to the new GDPR framework, operators will have to acquire permission from customers to use their personal data for service provisioning and will also need to explain how personal data is used.

An increasing number of end users is expected due to the directive which presumes that a large majority of households in EU will implement energy smart meters. The service requires near real time or real time sensitivity. It is offered at known locations and the generated data volume from a single meter is usually low. Billing is typically performed on a monthly basis by the utility company, while additional services may be charged per request. From all regulatory characteristics, privacy is the biggest concern since energy consumption can be used to infer user’s behavior. Therefore, a privacy policy has to be implemented according to the GDPR directive [25,26]. The major dependability attribute is high reliability requiring excellent continuous operational performance, while the main security requirement is integrity.

### 4.2. Smart Parking

According to [27], between 1927 and 2001, a number of studies of cruising in congested downtowns have found that drivers are spending on average between 3.5 and 14 min to find a parking space, and that between 8 and 74 % of the traffic in cities was generated by cruising for parking. To make traffic more fluid, cities are placing sensors at parking spots to detect available spaces. Traditionally, the information from the sensors is provided using LCD displays on the street, but nowadays mobile applications that offer real time information about available parking spaces are prevailing. In addition, such applications can also provide built-in navigation to guide you to an available parking space. One of the challenges for such applications is to prevent that someone else parks the car at a location allocated to you.

Due to its nature, Smart Parking services have a large number of end users, both citizens and visitors. The service has to provide information in real time about available parking spaces for a specific location. A mobile application is usually provided free of charge since the deployment and maintenance costs of a deployed solution can be covered from the parking fee. Also, additional services, e.g., reservation of parking spaces can be charged. A mobile application needs to provide GPS location of a user, and thus the service is location-based. Privacy policy for end users needs to be formulated and implemented according to the GDPR and Directive (EU) 2016/680 [25,26]. Security requirements in the context of Smart Parking are related to integrity, where user authenticity, accuracy and completeness of data and information related to the service are vital. One of the major challenging requirements is high service availability requiring excellent operational performance and maintenance which can ensure long mean time between failures and very short mean time to repair. Another challenge relates to providing an integrated parking service across different parking spaces and garages at the city level which may be operated by different stakeholders. In such environments, only interoperable parking solutions can offer the required flexibility and adaptability of the end user application to surrounding dynamic conditions of city-wide traffic. Open access to parking space information (i.e., their availability and location) is also vital to such integrated Smart Parking solutions. Thus, we can conclude that Smart Parking is a challenging service, both from the technical and regulatory perspective.

### 4.3. Mobile Crowd Sensing (MCS)

MCS is nowadays an emerging type of service which requires active citizen participation and engagement. The term crowd sensing describes services and applications “where individuals with sensing and computing devices collectively share data and extract information to measure and map phenomena of common interest” [28]. For example, such service can be used for air quality or noise monitoring in city streets to densely cover city area with measurements at specific micro locations which are hard to reach by official static measurement stations.

In case of air quality monitoring, an air quality wearable sensor is needed to act as an IoT device connected to a user’s smartphone where mobile application collects data from wearable sensors and sends them to cloud while citizens are moving through the city [29,30]. In case of noise monitoring, a smartphone’s microphone can be used to generate noise readings. Note that both sensors built into wearables and smartphones need to be calibrated to offer credible readings, while citizens need to be instructed on how to perform the measurements. Each measurement is tagged by a location generated by the smartphone’s GPS, thus it is location-based. The incentive to use the service is the possibility of data sharing within the community and collection of dense measurements about urban environments. An example mobile application for MCS named CUPUS Crowdsensing is available on Google Play (https://play.google.com/store/apps/details?id=hr.fer.tel.cupusmobileapp)—it targets cyclists and pedestrians as promising communities and early adopters of the service since they are highly affected by poor air quality in cities.

Considering that the service is based on measurements generated by individuals, the most interesting characteristics of this service are availability and integrity. Service availability depends entirely on volunteer’s willingness and motivation to collect measurements at all times; otherwise, the service cannot function properly. Integrity refers to data accuracy and consistency, and thus generated readings need to be constantly analyzed and monitored by experts to achieve and maintain service trustworthiness. Maintainability might pose a challenge since wearable sensors have to be periodically calibrated. The nature of the service is that crowds generate information and therefore sensors are with crowds. The question is whether crowds will calibrate sensors and use them correctly, so the measurement results may be questionable. Since citizen are generating and sharing the measurements, open access to the generated data which does not reveal any personal information is expected.

### 4.4. Smart Street Lighting

In today’s world reduction of electricity consumption is one of the most important topics for sustainable development of urban areas. Public lighting consumes a large amount of cities’ electricity consumption due to its continuous operation during night time. According to EC’s report [31] which is part of Digital Agenda for Europe [32], public lighting accounts for up to 60% of a typical municipality’s electricity costs. Also, legacy street lights are failure prone and costly to maintain, which increases lighting costs.

Smart or intelligent street lighting refers to a service where public street lighting adapts to movements of pedestrians, cyclists and cars. With embedded presence sensors and cameras, such solutions can in addition collect and transmit information that help cities monitor and respond to any safety-critical circumstances. Sensors and cameras may be used to detect traffic congestion or for Smart Parking application to track available parking spaces.

An expected number of end users is high, since all pedestrians and vehicles are service users. The service is provided in real time at fixed location, thus it is based on local interactions which reduce its complexity. Street light service as value added service for utility companies will not charge end users, however additional services may require payment. If detection of users based on face recognition features is accomplished using street cameras, privacy policy for end users needs to be is implemented according to GDPR and e-privacy directive [25,26]; otherwise no personal information is collected and privacy policy is not needed. A key security attribute is service integrity requiring accuracy and consistency, while from dependability attributes, reliability is the most important as a lighting system should function continuously. Operator switch is not important for the service since it is intended to be provided by utility companies. A key characteristic that is important for other services that want to reuse deployed presence sensors and cameras is interoperability, but strict access control needs to be enabled so that only authorized users have permission to access the information generated by those devices. Note that a street lamp may also host additional devices, like environmental sensors or mobile base stations (e.g., 5G, LoRaWAN, WiFi, Sigfox, etc.).

### 4.5. Results

All the regulatory and technical characteristics identified for the previous four use case scenarios are presented in Table 1. It compares the analyzed services according to the taxonomy introduced in Section 3.

Based on the comparison of Smart City services using the proposed taxonomy and roles explained in the IoT value chain model, we identify regulatory recommendations which are important for each role. Table 2 provides an overview of all regulatory aspects mapped to relevant stakeholders in the IoT value chain for the analyzed use cases.

To conclude, the most important regulatory characteristics for a device provider are interoperability at the level of device protocol stack and security issues, also at the device level, where data integrity and authorized access to device is of primary importance. An IoT connectivity provider is responsible to ensure device roaming (relevant only to MCS with wearable devices since devices used in other use cases are stationary) and lawful interception. The main concerns of an IoT platform provider refer to platform dependability and security. With respect of dependability, availability and reliability of services offered by the platform are the most important requirement of the analyzed services, while data integrity, accuracy and completeness as well as authorized access to services and corresponding data are the most important security-related requirements. Moreover, an IoT platform provider should have obligation to offer an interoperable solution with well-defined service interfaces so that provider switch is possible. It also needs to enable open data access in case of Smart Parking and MCS, and define adequate policies to protect personal data ensuring user privacy. Similarly to the IoT platform provider, the main concerns of an IoT service integrator relate to service security and user privacy. We expect that IoT service integrator can be changed in all use cases except in Smart Street Lighting. Application developer needs to guarantee data integrity within a developed application and ensure that only authenticated users have access to certain data and services. In case the application stores some personal data, appropriate policies need to be defined. Note that in case of Smart Street Lighting, we do not expect that a mobile application will be developed for end users. For an IoT user, privacy concerns are the most important, while an end user has no specific regulatory obligations except to rightfully report readings in case of MCS, and requires trust in the entire ecosystem when its personal data is collected, stored and processed.

One of the cities’ major problems is finding funds for the realization of their Smart City projects. We have witnessed the first wave of very successful research-driven pilots triggered by EU-funded projects where visionary administration and entrepreneurs have lead the way to city digitalization with a bottom-up innovation-driven approach regardless of national governments and national strategies. However, in environments which are lagging behind in terms of deployed infrastructure, the concept of Smart City development must follow a logical sequence of events and be in line with the development of required city infrastructure, including adequate communication infrastructure. Precisely for this reason, if we want to foster development of Smart Cities, one of the prerequisites is existence and preparation of a national strategy for the development of smart local environments or cities with an action plan that clearly defines the goals and how to achieve them. Each region (county or city) should prepare its own strategy that will follow guidelines from the national strategy while at the same time take into account specifics of the region. National and regional or local strategy requires strategic planning and implementation of corresponding initiatives and projects that will enable improvement of quality of life and will create new jobs for each specific region. It will also improve economic and social development with higher market competition. Smart City strategy should include a methodological framework and examples of good practices for the development of Smart City solutions and at the same time should explore those areas of application in which it is more effective and over which strategic goals are set. A similar coaching strategy is perused by the EU’s Digital Cities Challenge where selected cities (“Challenge cities”) are following the good examples of “Mentor cities” while aligning their solutions to specific regional and national strategies.

## 5. Conclusions

An accelerated growth in population of cities around the world is making continuous pressure on existing city infrastructure; therefore, city leaders are forced to make changes from traditional towards digital cities with strategies and plans for city development for the next 20-30 years. Also, technology is improving rapidly and legal framework is not adapted quickly to follow this pace of technical changes and innovations. Each year, governments around the world are trying to set the rules that would be appropriate for today’s technology and the main focus is to bring technology closer to people by raising awareness and trust in technology.

In contrast to existing literature which focuses on technical aspects of Smart City services, in this article we tried to analyze both technical and regulatory aspects of Smart City services to identify recommendations to city authorities, service developers and various stakeholders in the IoT value chain relevant to their commitments and obligations within an IoT ecosystem of Smart City services.

We proposed a taxonomy that categorizes and lists relevant technology and regulatory characteristics of Smart City services. The taxonomy was used to analyze four Smart City services, Smart Parking, Smart Metering, Mobile Crowd Sensing and Smart Street Lighting. Based on this model for each role in the IoT value chain specific recommendations have been given with regulatory aspects affecting their business. There are many technical and regulatory characteristics for Smart City service, however, each role in IoT value chain does not need to focus on and deal with all of them. Therefore, the identified recommendations simplify their business analyses. Although we have covered the most common legal and regulatory issues that are known today, designers should also take into consideration local laws and bylaws that might put some additional requirement on particular Smart City service deployments.

In future work we will focus on Value-Network-Analysis to explain the relationships between stakeholders in our IoT value chain model and to develop specific business model recommendations for selected use cases which take into account the regulatory aspects analyzed in this paper. Another question which arises is to identify the boundaries where responsibilities of one stakeholder stop and are taken over by another stakeholder, while legal responsibility of stakeholders in case of disputes and ownership of generated data still remains an open regulatory issue.

## Figures and Tables

**Figure 1 sensors-19-00415-f001:**
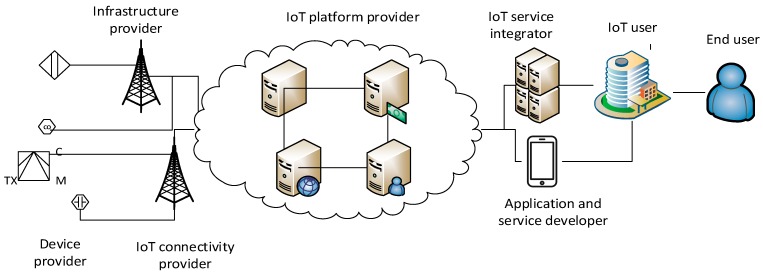
IoT value chain model.

**Figure 2 sensors-19-00415-f002:**
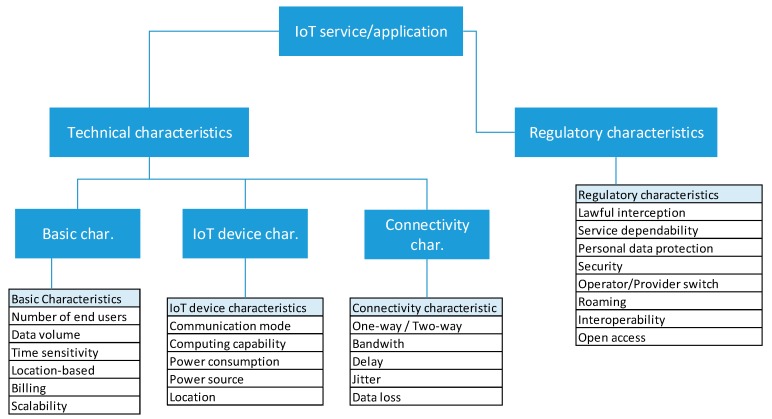
Taxonomy of Smart City service characteristics.

**Table 1 sensors-19-00415-t001:** Comparison of technical and regulatory characteristics of selected Smart City services.

Smart City Service	Smart Metering	Smart Parking	MCS	Smart Street Lighting
Basic characteristics				
No. of end users	High	High	Medium/high	High
Data volume	Low	Low	Low	Low/high
Time sensitivity	Near real time	Real time	Near real time, on demand	Real time
Location-based service	No	Yes	Yes	No
Billing	Yes	Yes	No	No
Scalability	Yes	Yes	Yes	Yes
Regulatory characteristics				
Lawful interception	No	Yes	Yes	No
Dependability	High reliability	High availability	Availability	Reliability
Privacy	Policy	Policy	Policy	Policy/not important
Security	Integrity	User authenticity, accuracy and completeness of data	Integrity	Integrity
Provider switch	Yes	Yes	Yes	Not important
Roaming	Not important	Not important	Yes	Not important
Interoperability	Policy	Integrated parking service	Policy	Policy
Open access	No	Yes	Yes	No
IoT device				
Access mode	Wireline or Wireless	Wireline or Wireless	Wireless	Wireline
Computing capability	No	No	No	yes
Power source	Main	Battery or main	Rechargeable battery	Main
Energy consumption	Small	Small	Small	Small
Location	Fixed	Fixed	Mobile	Fixed
Connectivity				
One-way/two-way	One-way/two-way	One-way (typically)	Two-way	Two-way
Bandwidth	Low	Low	Low	Low/medium
Delay	Best effort	Low (up to 5 sec)	Best effort	Best effort
Jitter	Not important	Not important	Not important	Not important
Data loss	Very low	Very low	Low	Very low

**Table 2 sensors-19-00415-t002:** Regulatory recommendations for each role in the IoT value chain.

		Analyzed Use Cases			
IoT Value Chain Model Role	Regulatory Characteristic	Smart Metering	Smart Parking	MCS	Smart Street Lighting
Device provider	Interoperability (protocol stack)	yes	yes	yes	yes
	Security (integrity and authorized access to device)	yes	yes	yes	yes
IoT connectivity provider	Roaming	no	no	yes	no
	Lawful interception	yes	yes	yes	yes
IoT platform provider	Interoperability (platform and service)	yes	yes	yes	yes
	Dependability	Reliability	High availability	Availability	Reliability
	IoT platform provider switch	yes	yes	yes	yes
	Security	Integrity	User authenticity, accuracy and Completeness of data	Integrity	Integrity
	Privacy	Policy	Policy	Policy	Not important
	Open Access	no	yes	yes	no
IoT service integrator	Service provider switch	yes	yes	yes	no
	Security	Integrity	User authenticity, accuracy and completeness of data	Integrity	Integrity
	Privacy	Policy	Policy	Policy	Policy/not important
Application developer	Security	Integrity, user authenticity	User authenticity, accuracy and completeness of data	Integrity	N/A
	Privacy	Policy	Policy	Policy/not important	N/A
IoT user	Privacy	yes	yes	yes	yes (if face recognition is applied)
End user	Trust (if user generates data)	yes	yes	yes	yes (if face recognition is applied)

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
