# Peer review of "A Regulatory View on Smart City Services"

_sensors, 2019, doi:10.3390/s19020415_

Reviewer 1 Report

Interoperability is one of the key topics that the authors underline, therefore it is suggested to mention and have a look at :

A. Brutti,P. de sabbata,A. Frascella,N. Gessa,R. Ianniello,C. Novelli,S. Pizzuti,G. Ponti, "Smart City Platform Specification: A Modular Approach to Achieve Interoperability in Smart Cities: Technology, Communications and Computing", January 2019, DOI: 10.1007/978-3-319-96550-5_2

In book: The Internet of Things for Smart Urban Ecosystems

Moreover, as regards Regulatory characteristics the authors should think and somehow stress the dependece on local laws (which can be low/medium/high). for instance, the solutions for smart metering and smart street lighting are highly dependent on national/local regulations.

this is somehow connected to a wider issue which is the one of replicability, if one solution is dependent on local regulations then it is not highly replicable.

authors should somehow cover this issue in the framework

Author Response

Dear Reviewer 1,

Responses to your comments can be found in the attached file.

Regards,

Mario Weber

Reviewer 2 Report

The paper "A Regulatory View on Smart City Services" offers to the reader a very interesting and innovative overview on an emerging topic related to the smart city transition process.

The suggested taxonomy results quite innovative in helping local governments and stakeholders in the process of identifying key characteristics of their future smart city services.

Identifying in advance whether or not a specific service may be in line with regulatory and technical requirements for its successful deployment and usage in practice is of paramount interest in a so complex and rapidly changing scenario.

I would recommend the authors to:

add references (at least the website) to the inspiring examples they provide (e.g. smart app in Bologna, smart parking in Split), to let readers easily experience how these services look like in the reality;

check typos or inconsistencies in punctuation (brackets, .., lists…);

remove tables from 2 to 5 and consequently revise the text. Table 6 will be sufficient in showing main findings by adding 4 columns on the right (one for each service/example) and flagging single cells where characteristics are relevant for the related service.

I am sorry to say that I disagree on your comments from line 608 to 621, because the lack of national or top-down smart city strategies may be or not an element slowing down the process of smart city development. However, most promising smart city projects supported by H2020 funding are bottom-up driven and spreading around Europe independently from national governments. These projects have been settled up but thanks to innovative local administrations and visionary researchers/entrepreneurs. See on this point some works done by Mosannenzadeh and colleagues in the smart city – smart energy city. I do not have evidence that national strategies will effectively help cities in developing their own solutions. On the other side, reinforcing international cooperation thanks to EIP platforms or similar seems working well.

Author Response

Dear Reviewer 2,

Responses to your comments can be found in the attached file.

Regards,

Authors

Reviewer 3 Report

Authors in this article made an effort to model Smart City Services based on relevant technologies and regulatory characteristics. Moreover, they highlight how the proposed model can be used or applied in a series of use cases like smart {parking, metering, streelighting} and urban crowd sensing and Smart.

Despite the fact, that the paper is quite well-elaborated and pleasant to read the overall contributions are not very clear (to me):

the discussion highlights the quite complex ecosystem and context constraints (technical, legal, economical etc.) in designing, building, deploying, maintaining a smart city application/service but this is something that has repetitively reported in a series of smart city EU-research project and initiatives like SmartSantanter (http://www.smartsantander.eu/), Organicity (https://organicity.eu/), Synchronicity (https://synchronicity-iot.eu/), Open & Agile Smart Cities (https://oascities.org)   and https://smartcities-infosystem.eu/

The intended audience lies more on business or governance side in contract to the sensor network, future internet, iot research focus of sensors mdpi. An SME, for example, is not so clear how can benefit from the overall discussion.

The idea of designing a cost-benefit model  (or sustainability) of smart city apps/services (that will factor out and describe investment efforts -technical, socio-economical aspects -and will report against socio-economical benefits gains) is quite interesting and challenging at the same time. Paper seems that touched bits-and-pieces of the aspect.

Author Response

Dear Reviewer 3,

Responses to your comments can be found in the attached file.

Regards,

Authors

Round  2

Reviewer 3 Report

Authors tackled all of the comments adequately. Moreover, they improved sections 1 and 4 significantly.